# On the Hydration of Heavy Rare Earth Ions: Ho^3+^, Er^3+^, Tm^3+^, Yb^3+^ and Lu^3+^—A Raman Study

**DOI:** 10.3390/molecules24101953

**Published:** 2019-05-21

**Authors:** Wolfram Rudolph, Gert Irmer

**Affiliations:** 1TU Dresden, Medizinische Fakultät, Institut für Virologie im MTZ, Fiedlerstr. 42, 01307 Dresden, Germany; 2Technische Universität Bergakademie Freiberg, Institut für Theoretische Physik, Leipziger Str. 23, 09596 Freiberg, Germany; irmer@physik.tu-freiberg.de

**Keywords:** Raman spectroscopy, aqueous Lu(ClO_4_)_3_-, Yb(ClO_4_)_3_-, Tm(ClO_4_)_3_-, Er(ClO_4_)_3_- and Ho(ClO_4_)_3_- solutions, YbCl_3_-solutions, ν_1_ Ln–O breathing modes, Yb^3+^- chloro-complex species

## Abstract

Raman spectra of aqueous Ho^3+^, Er^3+^, Tm^3+^, Yb^3+^, and Lu^3+^-perchlorate solutions were measured over a large wavenumber range from 50–4180 cm^−1^. In the low wavenumber range (terahertz region), strongly polarized Raman bands were detected at 387 cm^−1^, 389 cm^−1^, 391 cm^−1^, 394 cm^−1^, and 396 cm^−1^, respectively, which are fairly broad (full widths at half height at ~52 cm^−1^). These isotropic Raman bands were assigned to the breathing modes, ν_1_ Ln–O of the heavy rare earth (HRE) octaaqua ions, [Ln(H_2_O)_8_]^3+^. The strong polarization of these bands (depolarization degree ~0) reveals their totally symmetric character. The vibrational isotope effect was measured in Yb(ClO_4_)_3_ solutions in H_2_O and D_2_O and the shift of the ν_1_ mode in changing from H_2_O to D_2_O further supports the character of the band. The Ln–O bond distances of these HRE ions (Ho^3+^, Er^3+^, Tm^3+^, Yb^3+^, and Lu^3+^) follow the order of Ho–O > Er–O > Tm–O > Yb–O > Lu–O which correlates inversely with the band positions of the breathing modes of their corresponding octaaqua ions [Ln(OH_2_)_8_]^3+^. Furthermore, the force constants, *k_Ln–O_*, were calculated for these symmetric stretching modes. Ytterbium perchlorate solutions were measured over a broad concentration range, from 0.240 mol·L^−1^ to 2.423 mol·L^−1^, and it was shown that with increasing solute concentration outer-sphere ion pairs and contact ion pairs were formed. At the dilute solution state (~0.3 mol·L^−1^), the fully hydrated ions [Yb(H_2_O)_8_]^3+^ exist, while at higher concentrations (C_T_ > 2 mol·L^−1^), ion pairs are formed. The concentration behavior of Yb(ClO_4_)_3_ (aq) shows similar behavior to the one observed for La(ClO_4_)_3_(aq), Ce(ClO_4_)_3_(aq) and Lu(ClO_4_)_3_(aq) solutions. In ytterbium chloride solutions in water and heavy water, representative for the behavior of the other HRE ions, 1:1 chloro-complex formation was detected over the concentration range from 0.422–3.224 mol·L^−1^. The 1:1 chloro-complex in YbCl_3_(aq) is very weak, diminishing rapidly with dilution and vanishing at a concentration < 0.4 mol·L^−1^.

## 1. Introduction

In aqueous solution, the HRE ions of holmium, erbium, thulium, ytterbium, and lutetium exist in the tervalent state [1] and, with their high charge to radius ratio, are strongly hydrated [2,3]. The HRE ions possess eight water molecules arranged in a square antiprismatic geometry (S_8_ symmetry) in their first coordination sphere. The hydration geometry of heavy rare earth ions in aqueous solution was determined by X-ray (XRD) and neutron diffraction (ND) [4,5,6] as well as extended X-ray absorption fine structure (EXAFS) [7,8,9,10] techniques. Computer simulations contributed to clarifying the details of the structure and dynamics of the waters in the first hydration sphere of the HRE ions [11,12,13] and their results confirmed the octahedral coordination.

While XRD and neutron diffraction measurements [4,5,6] are carried out in concentrated solutions, EXAFS spectra [7,8,9,10] are measured on dilute or moderately concentrated solutions. At high concentrations, however, it is known that the Ln^3+^(aq) ions form complex species/ion pairs with common ions (Cl^−^, NO_3_^−^ and SO_4_^2−^). Perchlorate, however, acts as a weak complex forming anion and the anion is therefore suitable for hydration studies.

Raman spectroscopy has been applied frequently to characterize hydrated metal ions and related species in aqueous solution and is especially useful in characterizing the species formed at the molecular level such as hydrated ions, ion pairs between metal ions and anions or metal ion hydrolysis. Raman measurements on aqueous Ln^3+^(aq) ions should allow, in principle, the characterization of the solution structure in greater detail. Recently, light rare earth solutions (La^3+^, Ce^3+^, Pr^3+^, Nd^3+^ and Sm^3+^) as well as the solution of the HRE ion, Lu^3+^, have been measured using Raman spectroscopy [14,15,16,17] and it has been shown that the application of the so-called R-procedure is crucial in exploring the low frequency part of the Raman scattering.

The present study was undertaken to characterize the hydration and speciation in aqueous Ho^3+^, Er^3+^, Tm^3+^, and Yb^3+^ perchlorate solutions. Lu^3+^- solutions in water and heavy water have been measured and described in a recent publication [17] and these results are also included in this work. The Ho^3+^-, Er^3+^-, Tm^3+^- and Yb^3+^- perchlorate solutions were measured over broader concentration ranges and down to the terahertz frequency region at 22 °C. Furthermore, Yb(ClO_4_)_3_ solutions in water and heavy water were measured in order to characterize the vibrational isotope effect changing from [Yb(H_2_O)_8_]^3+^ to [Yb(D_2_O)_8_]^3+^ and the symmetric stretching modes of the two species were observed.

In chloride solutions, however, it was shown that light rare earth ions form chloro-complex species, [14,15,16,17]. In order to clarify whether chloro-complex species may also form in these HRE chloride solutions, YbCl_3_ solutions were measured at varying concentrations in water and heavy water. Recently, inner-sphere chloro- complexes were detected in aqueous solutions on a variety of rare earth chloride solutions using Raman spectroscopy [14,15,16,17].

In this study, we are specifically interested in the vibrational characterization of the Ln–O stretching modes of the octahydrates (fully hydrated ions, [Ln(OH_2_)_8_]^3+^) in dilute perchlorate solutions. The formation of ion pairs between Yb^3+^ ions and perchlorate has been studied on a concentration series of Yb(ClO_4_)_3_ solutions representative for other HRE ions. The influence of Cl^−^ on the fully hydrated Ln^3+^(aq) was studied exemplarily on YbCl_3_ solutions in water and heavy water. These results are discussed in connection with recently measured aqueous LuCl_3_ solutions [17]. 

## 2. Experimental Details and Data Analysis 

Preparation of Solutions

The rare earth ion concentrations of all solutions were analysed by complexometric titration [18]. The solution densities were determined pycnometrically at 22 °C and the molar ratios of water per salt calculated (R_w_ -values). The solution pH values were measured with pH meter S220 using a pH electrode InLab Expert Pro-ISM (Mettler –Toledo GmbH, Deutschland, Giessen). For Raman spectroscopic measurements, the solutions were filtered through a fine sintered glass frit (1–1.6 µm pore size). The preparation of the lutetium perchlorate solutions at various concentrations were described earlier [17]. 

Ytterbium perchlorate solutions were prepared from Yb(ClO_4_)_3_∙6H_2_O (Alfa-Aesar (Thermo Fisher), 99.9%, Kandel, Deutschland) dissolved with ultrapure water (PureLab Plus, Ultra-pure Water Purification Systems). A concentrated Yb(ClO_4_)_3_ solution was prepared at 2.423 mol·L^−1^ (R_w_ = 16.86). This solution was acidified with a slight amount of HClO_4_ and a pH value at ~ 2.0 was obtained. From this stock solution, the following dilution series was prepared: 1.217 mol·L^−1^ (R_w_ = 39.30), 0.808 mol·L^−1^ (R_w_ = 62.13), 0.603 mol·L^−1^ (R_w_ = 85.16) and 0.240 mol·L^−1^ (R_w_ = 223.6). The solutions were analysed for dissolved chloride with a 5% AgNO_3_ solution and the absence of a white AgCl precipitate was proof that the stock solution was free of Cl.

Two Yb(ClO_4_)_3_ solutions in heavy water were prepared from a 2.526 mol·L^−1^ Yb(ClO_4_)_3_(D_2_O) stock solution with heavy water (99.9 atom% D; Sigma-Aldrich now Merck KGaA, Darmstadt, Germany) at 1.276 mol·L^−1^ and 0.779 mol·L^−1^. The deuteration degree for the solution was determined at ~97% D.

YbCl_3_ solutions were prepared from YbCl_3_∙6H_2_O (Sigma-Aldrich, 99.5%) and ultrapure water (PureLab Plus, Ultra-pure Water Purification Systems, now ELGA Labwater, Celle, Deutschland) by weight at 3.224 mol·L^−1^ (R_w_ = 15.64), 1.600 mol·L^−1^ (R_w_ = 33.06), 0.802 mol·L^−1^ (R_w_ = 67.55) and 0.400 mol·L^−1^ (R_w_ = 137).

A 3.300 mol·L^−1^ YbCl_3_ stock solution in D_2_O was used to prepare two dilute YbCl_3_ solutions with heavy water (99.9 atom% D; Sigma-Aldrich) at 0.844 mol·L^−1^ and 0.422 mol·L^−1^. The deuteration degree in the dilute solutions was determined at ~96.5% D. 

A Tm(ClO_4_)_3_ solution was prepared from Tm_2_O_3_ (Sigma-Aldrich, 99.9%) which was dissolved with 6 mol·L^−1^ HClO_4_ solution (Fisher Scientific GmbH, Schwerte, Deutschland until a clear solution was obtained. The solute concentration was determined at 1.897 mol·L^−1^ and two dilute solutions were prepared from the stock solution and triply distilled water by weight at 0.980 mol·L^−1^ and 0.315 mol·L^−1^. These solutions contained a slight excess of HClO_4_.

An Er(ClO_4_)_3_ stock solution was a commercial product from Alfa-Aesar (Thermo Fisher) (Kandel, Deutschland) at 50 wt%, Reagent Grade (99.9%) at 2.245 moL^−1^. Two dilute solutions at 1.123 moL^−1^ and at 0.321 moL^−1^ were prepared by weight with ultrapure water. These solutions contained a slight excess of HClO_4_ (pH value ~1.5).

A Ho(ClO_4_)_3_ stock solution was prepared from Ho_2_O_3_ (Sigma-Aldrich, 99.9%) which was dissolved with 6 mol·L^−1^ HClO_4_ (Riedel-de Haen, 70 wt%) until a clear solution was obtained. The solute concentration was determined at 1.675 mol·L^−1^. Two dilute solutions at 0.838 mol·L^−1^ and 0.240 mol·L^−1^ were prepared by weight from the stock solution with ultrapure water. The solutions contained a slight excess of HClO_4_ (pH value ~1.75). 

Raman spectroscopic measurements have been reported in detail elsewhere, so only a brief summary is given [19,20]. Raman spectra were measured in the macro chamber of the T 64000 Raman spectrometer from Jobin Yvon in a 90° scattering geometry at 22 °C. A quartz cuvette was used (Hellma Analytics, Müllheim, Germany) with a 10 mm path length and a volume at 1000 µL. The spectra were excited with the 487.987 nm or the 514.532 nm line of an Ar^+^ laser at a power level of 1100 mW at the sample. The Yb^3+^-perchlorate and -chloride solutions have no visible absorption bands and therefore, both excitation wavelengths may be used. Tm(ClO_4_)_3_ solutions were measured with the 514.532 nm Ar^+^ line. Er(ClO_4_)_3_ solutions were excited with the 487.987 nm Ar^+^ line and only the most dilute solution could be measured reliably because the concentrated ones were strongly absorbing at the absorption gap at ~488 nm of its UV-vis spectrum. Ho(ClO_4_)_3_ solutions were excited with the 514.532 nm Ar^+^ line and only the most dilute solution could be reliably measured.

After passing the spectrometer in subtractive mode, with gratings of 1800 grooves/mm, the scattered light was detected with a cooled CCD detector. The scattering geometries I_VV_ = (X[ZZ]Y) and I_VH_ = (X[ZX]Y) are defined as follows: the propagation (wave vector direction) of the exciting laser beam is in X direction and the propagation of the observed scattered light is in Y direction, the 90° geometry. The polarisation (electrical field vector) of the laser beam is fixed in Z direction (vertical) and the polarisation of the observed scattered light is observed in Z direction (vertical) for the I_VV_ scattering geometry. For I_VH_ the electric field vector of the exciting laser beam is in Z direction (vertical) and the observed scattering light is polarized in the X direction (horizontal). Thus, for the two scattering geometries it follows:(1)$$I_{\mathrm{VV}}=I(X[\mathrm{ZZ}]Y)\;=\;45{{\bar{\alpha }}^{\prime }}^{2}+\;4{\gamma^{\prime }}^{2},$$
(2)$$I_{\mathrm{VH}}=\;I(X[\mathrm{ZX}]Y)=\;3{\gamma^{\prime }}^{2},$$

The symbols ${\bar{\alpha }}^{'}$ and γ’ are the isotropic and the anisotropic invariant of the Raman polarizability tensor, respectively [19]. The isotropic spectrum (I_iso_) was constructed according to Equation (3):
I_iso_ = I_VV_ − 4/3I_VH_,(3)

The polarization degree of the Raman bands, ρ (ρ = I_VH_/I_VV_) was determined using an analyzer and adjusted, if necessary, before each measuring cycle using CCl_4_ [19].

The calibration of the Raman spectra has been carried out using plasma lines [19]. The accuracy of the peak positions for the perchlorate deformation modes were not better than ±0.5 cm^−1^ and for the much narrower ν_1_(a_1_)ClO_4_^−^ band ±0.2 cm^−1^. The peak positions of the bands were determined by fitting the baseline corrected bands with a Gauss-Lorenz product function (see ref. [20]). The accuracy of the weak and much broader ν_1_ Ln–OH_2_/OD_2_ bands was ±1 cm^−1^ using the perchlorate band at 461 cm^−1^ as an internal reference band.

In order to characterize the spectral features in the low wave number region, the Raman spectra in I-format were reduced and the R-spectra obtained. The R($\tilde{\nu }$) spectra are independent of the excitation wavenumber ${\tilde{\nu }}_{L}$ and the measured Stokes intensity is further corrected for the scattering factor (ν_L_-$\tilde{\nu }$)^3^. (The scattering factor must be to the power of 3 when applying counting methods [21].) The spectra were further corrected for the Bose-Einstein temperature factor, B = [1-exp(-h$\tilde{\nu }$c/kT)] and the frequency factor,$\tilde{\nu }$, to give the so-called reduced spectrum, R($\tilde{\nu }$) (detailed in earlier publications [19,20]). The isotropic spectrum in R-format, R_iso_, is calculated according to equation 3 but substituting the spectra in I-format, I_VV_ and I_VH_ with R_VV_ and R_VH_. In the low wavenumber region, the I($\bar{v}$) and R($\bar{v}$) spectra are significantly different and only the spectra in R-format are presented. An advantage in applying isotropic R-spectra is the almost flat baseline in the terahertz region allowing relatively unperturbed observation of any weak modes present.

## 3. Results and Discussion

### 3.1. Raman Spectra on Aqueous Ln(ClO_4_)_3_ Solutions (Ln = Lu^3+^, Yb^3+^, Tm^3+^, Er^3+^ and Ho^3+^)

The perchlorate spectrum, ClO_4_^−^(aq): The Raman spectrum of the perchlorate ion in aqueous solution (NaClO_4_(aq)) has been well characterized and only a brief description shall be given [14,15,16]. The ClO_4_^−^ ion possesses T_d_ symmetry and has nine modes of internal vibrations spanning the representation Γ_vib_(T_d_) = a_1_(Ra) + e(Ra) + 2f_2_(Ra, i.r.). All normal modes are Raman active, but in i.r. only the f_2_ modes are allowed. The ν_1_(a_1_) ClO_4_^−^ band, centred at 931.5 cm^−1^, is totally polarized (ρ = 0.005) whereas ν_3_(f_2_) ClO_4_^−^, centred at 1108 cm^−1^, is depolarized. The deformation modes ν_4_(f_2_) ClO_4_^−^ at 629 cm^−1^ and ν_2_(e) ClO_4_^−^ at 461 cm^−1^ [15] are also depolarized. In dilute aqueous NaClO_4_ solutions, the spectrum of ClO_4_^−^ (aq) shows no sign of contact ion pairs or outer-sphere ion pairs; the ν_1_(a_1_) ClO_4_^−^ band at 931.5 cm^−1^ is quite narrow with a full width at half height (fwhh) = 7.2 cm^−1^. However, the ν_1_(a_1_) ClO_4_^−^ band shows an intrinsic low frequency shoulder at 923 cm^−1^ which is caused by Fermi resonance of the overtone of ν_2_(e) ClO_4_^−^, 2x ν_2_(e) = 923 cm^−1^ with ν_1_(a_1_) ClO_4_^−^ band [15]. The antisymmetric stretching mode, ν_3_(f_2_) ClO_4_^−^, is much weaker in intensity than ν_1_(a_1_) and it appears at 1106 cm^−1^ with a fwhh = 65 cm^−1^. In Figure 1A the scattering profiles in R-format (R_VV_, R_VH_, and R_iso_) of a 3.801 mol·L^−1^ NaClO_4_(aq) are presented from 50 to 800 cm^−1^ and in addition the overview Raman scattering profiles (I_VV_, I_VH_ and I_iso_) from 50 to 1800 cm^−1^ are given in Appendix A together with the band positions and assignments.

The Raman spectra of Lu(ClO_4_)_3_(aq), Yb(ClO_4_)_3_(aq), Tm(ClO_4_)_3_(aq), Er(ClO_4_)_3_(aq) and Ho(ClO_4_)_3_(aq): First, we present and discuss the spectroscopic results on Yb(ClO_4_)_3_ followed below by the spectra of aqueous Tm(ClO_4_)_3_, Er(ClO_4_)_3_ and Ho(ClO_4_)_3_ solutions. The Raman spectra of aqueous Lu(ClO_4_)_3_ solutions have been reported recently [17] with the ν_1_ LuO_8_ breathing mode at 396 cm^−1^.

Figure 1B,C show the scattering profiles of aqueous Yb(ClO_4_)_3_ solutions at 0.240 mol·L^−1^ (Rw = 226.6) and, in comparison, a solution at 2.423 mol·L^−1^ (R_W_ = 16.86) respectively. The Raman scattering profiles of the 0.240 mol·L^−1^ aqueous Yb(ClO_4_)_3_ solution (Figure 1B) from 40–750 cm^−1^ shows two ClO_4_^−^(aq) bands at 461 cm^−1^ and 629 cm^−1^ and a broad, weak polarized mode (isotropic scattering) at 394 cm^−1^ which does not occur in NaClO_4_(aq). Therefore, the band at 394 cm^−1^ has to be assigned to the ν_1_ YbO_8_ breathing mode of the [Yb(H_2_O)_8_]^3+^ species. In the at 2.423 mol·L^−1^ (R_W_ = 16.86) Yb(ClO_4_)_3_ solution (Figure C), however, the ν_1_ YbO_8_ breathing mode is shifted by 4 cm^−1^ to lower wavenumbers compared to the dilute solution (Figure 1B). An overview Raman spectrum of the 2.423 mol·L^−1^ Yb(ClO_4_)_3_ solution is given in Appendix A (top panel) from 80–1400 cm^−1^ which displays all four perchlorate bands and the weak isotropic mode at 390 cm^−1^ assigned to the ν_1_ YbO_8_ breathing mode. The vibrational bands in the anisotropic scattering could only be detected in the concentrated Yb(ClO_4_)_3_ solution at 2.423 mol·L^−1^ because of their very weak and broad nature. These anisotropic bands are even weaker than the already weak ν_1_ YbO_8_ breathing mode with an integrated band intensity at 3161. The band fit results for the anisotropic scattering are given in Appendix A and presented in Figure 2 Five bands appear at 88.5 cm^−1^ (fwhh = 119), 158.7 cm^−1^ (fwhh = 97.6.9 cm^−1^), 229.4 cm^−1^ (fwhh = 75.7 cm^−1^), 260.0 cm^−1^ (fwhh = 68cm^−1^) and 333.2 cm^−1^ (fwhh = 85.3) in the anisotropic scattering in the 2.423 mol·L^−1^ Yb(ClO_4_)_3_(aq) solution. These weak, broad bands stem from the YbO_8_ skeleton fundamentals of the [Yb(OH_2_)_8_]^3+^ species and break the symmetry of the YbO_8_ skeleton. Therefore, they appear only in the anisotropic scattering, but not in the isotropic profile. From group theoretical considerations we expect 7 Raman active modes for the YbO_8_ skeleton (ligated water molecules seen as point masses) and a brief group theoretical discussion shall be given. The YbO_8_ skeleton (D_4d_ symmetry) with its 9 atoms leads to 21 normal modes and the irreducible representation follows as: Γ_v_(D_4d_) = 2a_1_(Ra) + b_1_(i.a.) + 2b_2_(i.r.) + 3e_1_(i.r.) + 3e_2_(Ra) + 2e_3_(Ra). (The YbO_8_ skeleton possesses no symmetry centre but the mutual exclusion rule nevertheless applies.) Seven modes with the character a_1_, e_2_ and e_3_ are Raman allowed while six modes with the character b_1_, b_2_ and e_1_ are i.r. active. The totally symmetric Yb–O stretch, the breathing mode, is only Raman active and appears strongly polarized in the Raman spectrum as the strongest band of the YbO_8_ skeleton. Two additional depolarized Raman stretching modes are expected (character e_2_ and e_3_) as well as four other Raman deformation modes (character a_1_, e_2_ and e_3_). In infrared, two stretching modes (character b_2_ and e_1_) are expected and the remaining are deformations. In reality, however, we observe only six skeleton modes with one unaccounted mode (see also [17]). From our Raman spectroscopic results, it follows directly that the Yb^3+^–OH_2_ hydration shell cannot constitute a hexa-hydrate (T_h_ symmetry) which has been, for instance, characterized for [Al(OH_2_)_6_]^3+^(aq) [22,23]. Group theoretical considerations expect only three skeleton modes in Raman for [Al(OH_2_)_6_]^3+^; one of which should be totally polarized (breathing mode for the AlO_6_ skeleton) and the remaining two depolarized. All of these bands were detected in the Raman spectrum of an Al(ClO_4_)_3_(aq) solution with the symmetric stretching mode of [Al(OH_2_)_6_]^3+^ at 525 cm^−1^ strongly polarized and two bands at 438 cm^−1^ and 332 cm^−1^ which are depolarized [22,23].

The concentration dependence of the band parameters (peak positions, full width at half height (fwhh) and integrated band areas) of the ν_1_ YbO_8_ breathing mode for Yb(ClO_4_)_3_ solutions allows the determination of the change of these band parameters as a function of concentration. The band profiles of the ν_1_ YbO_8_ breathing mode are given in Figure 3 at concentrations 0.240 mol·L^−1^ (R_W_ = 268.74), 0.603 mol·L^−1^ (R_W_ = 85.17), 0.808 mol·L^−1^ (R_W_ = 62.13), 1.217 mol·L^−1^ (R_W_ = 16.86) and finally at 2.423 mol·L^−1^ (R_W_ = 16.86). The ν_1_Yb–O breathing mode appears at 394 cm^−1^ at the lowest concentrations and shifts ~4 cm^−1^ to lower wavenumbers at the highest concentration. Furthermore, the bandwidths also increase with increasing solute concentration from 52 cm^−1^ for the 0.240 mol·L^−1^ solution to 59 cm^−1^ for the 2.423 mol·L^−1^ solution (Figure 3). This slight change in band parameters of ν_1_ YbO_8_ breathing mode with increasing solute concentration may be due to ion pair formation in concentrated solutions (The ion pairing effect in perchlorate solutions is discussed in detail in [15,17]). The integrated band intensity of the ν_1_YbO_8_ breathing mode, A_394_, rises linearly with the solution concentration. The dependence of the integrated band intensity of the ν_1_ YbO_8_ breathing mode of [Yb(OH_2_)_8_]^3+^ as a function of the Yb(ClO_4_)_3_ concentration is given in Appendix A and for the linear relationship follows: A_394_ = 1303.7·C_Yb(ClO4)3_ (R^2^ = 99.9).

In addition to the ν_1_YbO_8_ of [Yb(OH_2_)_8_]^3+^ an extremely weak and broad band centered at 170 ± 10 cm^−1^ appears isotropic Raman scattering in of aqueous Yb(ClO_4_)_3_ solution (see Appendix A, top panel). This isotropic band is assigned to a restricted translational mode of the weakly H- bonded water molecules (O-H∙∙∙∙OClO_3_^−^). The mode is strongly anion and concentration-dependent [14,15,16]. The influence of the ClO_4_^−^ on the water spectrum has been discussed in recent studies on aqueous Ln(ClO_4_)_3_ solutions [14,15,16,17]. The Raman scattering profiles, I_VV_, I_VH_ and I_iso_ of the O-H stretching region of H_2_O and its bending mode of a Yb(ClO_4_)_3_ solution at 2.423 mol·L^−1^ and their peak positions are given in Appendix A, bottom panel, and for details see [14,15,16].

The triflate ion (trifluoromethanesulfonate) in aqueous solution acts as an even weaker complex forming anion and is suited, therefore, for studying metal ion hydration. In aqueous solution, however, the weak ν_1_ band of [Yb(OH_2_)_8_]^3+^ at ~394 cm^−1^ is overlapped by a strongly polarized triflate band at 319 cm^−1^ and so band fit analysis was applied. The isotropic Raman spectrum of Yb(CF_3_SO_3_)_3_(aq) at 1.25 mol·L^−1^ is shown in Appendix A and the band fit analysis gave two bands with the first band component at 319 cm^−1^ and the second band at 394 cm^−1^ (fwhh = 50 cm^−1^). The first band, a polarized band, stems from CF_3_SO_3_^−^ (aq) but the second band is the ν_1_ YbO_8_ breathing mode of [Yb(H_2_O)_8_]^3+^. Band parameters and assignments of CF_3_SO_3_^−^(aq) modes are given in [16]. 

The effect of deuteration on the YbO_8_ skeleton modes of [Yb(OD_2_)_8_]^3+^ was studied in Yb(ClO_4_)^3−^ D_2_O solution and resulted in a shift of the Yb–OD_2_ mode down to 374 cm^−1^. The shift of ν_1_ on deuteration is given as ν _1_’ = ν _1_[m(H_2_O)/m(D_2_O)]^1/2^ = $(394{\mathrm{cm}}^{-1})\cdot \sqrt{18.02/20.03}$ = 0.9485 × 394 cm^−1^ = 373.7 cm^−1^. (The water and heavy water molecules are taken as point masses.) A Raman spectra of Yb(ClO_4_)_3_ solutions in D_2_O at 0.779 mol·L^−1^ is presented in Figure 4 and for a concentration at 1.276 mol·L^−1^ in Appendix A. This isotope shift of the symmetrical stretch of YbO_8_ in changing from [Yb(OH_2_)_8_]^3+^(H_2_O) to [Yb(OD_2_)_8_]^3+^(D_2_O) and the totally symmetric character of the mode, that is, showing a polarization degree ~0, are indeed proof for the character of this mode. 

In the Raman spectra of the Tm(ClO_4_)_3_, Er(ClO_4_)_3_ and Ho(ClO_4_)_3_ solutions appear also as strongly polarized bands and were observed at 391, 389 cm^−1^, and 387 cm^−1^ respectively. These isotropic bands are unique in these HRE ion solutions and cannot be found in the hydrated ClO_4_^−^ (aq) spectrum.

The representative Raman spectra of Tm(ClO_4_)_3_, Er(ClO_4_)_3_ and Ho(ClO_4_)_3_ solutions are given in Appendix A, respectively. Several concentrations of the Tm(ClO_4_)_3_(aq) were measured but from the coloured Er(ClO_4_)_3_ and Ho(ClO_4_)_3_ solutions only the dilute solutions could be reliably measured. (Concentrated Er(ClO_4_)_3_ and Ho(ClO_4_)_3_ solutions absorb the laser light markedly.) The peak positions for the ν_1_ LnO_8_ breathing modes for [Lu(OH_2_)_8_]^3+^ (taken from [17]), [Yb(OH_2_)_8_]^3+^, [Tm(OH_2_)_8_]^3+^, [Er(OH_2_)_8_]^3+^ and [Ho(OH_2_)_8_]^3+^ are given in Table 1. Force constant calculations for the ν_1_ LnO_8_ breathing modes of this species, applying a simple model, have been carried out according to equation (4):(4)$$k_{M-O}={4\pi }^{2}c^{2}{\tilde{\nu }}_{i}^{2}N^{-1}A_{L},$$
with c, the velocity of light, ${\tilde{\nu }}_{i}$ the wavenumber of the mode *i*, N the Avogadro constant and A_L_ the molecular weight of the ligand, in our case water. The force constants, *k*_Ln–O_, calculated for the measured ν_1_ breathing modes are given in Table 1 together with the corresponding Ln^3+^- O bond distances [7]_._ The force constants increase from Lu^3+^, Yb^3+^, Tm^3+^, Er^3+^ and Ho^3+^ in the same order as the corresponding Ln–O bond distances decrease, namely Lu–O < Yb–O < Tm–O < Er–O < Ho–O (Appendix A).

Relative scattering intensities, S_h_, for the ν_1_ Ln–O breathing modes are also given in Table 1 and for the definitions of the S_h_ see ref. [20]. The small scattering intensity values at 0.0156 to 0.0165 for the ν_1_ Ln–O modes of the HRE octahydrates reflect the fact that the Ln–OH_2_ bonds possess low polarizability and are hard cations [25]. The accuracy of the scattering coefficient is not better than ± 0.0004 due to the low scattering intensity, the broadness of the modes and the uncertainties in subtracting the baseline. (Note that the S_h_ value for the totally symmetric stretching mode, ν_1_Lu–O is 0.0156 ± 0.0004 and the value reported in [17] is too small.)

From ab initio quantum mechanical charge field molecular dynamics studies, the mean Ln–O bond distances (Ln = Ho^3+^, Er^3+^, Tm^3+^, Yb^3+^ and Lu^3+)^ of the octahydrates, [Ln(OH_2_)_8_]^3+^, average coordination numbers, vibrational frequencies and the corresponding force constants were presented [13]. The authors claimed an “excellent agreement with experimental results” [13] of the computed frequencies with the measured ones in the glassy state [26]. The theoretical force constants for the ν_1_ breathing modes in [13] deviate considerably from our data in Table 1and do not follow the expected trend given in Appendix A in going from holmium to lutetium. This trend reflects the steady increase of the force constants of the Ln–O breathing modes with decreasing Ln–O bond distances in going from holmium to lutetium (Table 1; Appendix A). The force constant for ν_1_ Er–OH_2_ breathing mode in [13] was given 360 cm^−1^ equal to the one for the ν_1_ La–OH_2_ breathing mode of [LaOH_2_)_9_]^3+^. However, our recently published datum for the ν_1_ La–OH_2_ breathing mode [14,15] is with 343 cm^−1^ much smaller. The calculations in [13] are based on the simplified model of a heteronuclear diatomic species but such an assumption may not be correct. The character of the symmetrical normal mode ν_1_ of the LnO_8_ skeleton of the corresponding [Ln(OH_2_)_8_]^3+^ species reveals that the central cations remain stationary and only the water molecules are involved in the breathing motion without disturbing the symmetry and therefore these normal modes are totally polarized.

It is known from kinetic studies [24,27,28] that the water exchange reactions of the [Ln(OH_2_)_8_]^3+^ species for the octahydrates are very fast and these ions are known to be labile. From the rate constants, *k*ex at 25 °C, given in Table 1, follow the water residence times, the time the water molecules reside at these cations. The water residence times are in the range of several nanoseconds (see Table 1) which shows that these ions are indeed quite labile. From the vibration periods of the ν_1_ Lu–O modes which are at 0.086 to 0.084 ps in going from holmium(III) to lutetium(III) it follows that these species vibrate several hundred thousand times [17] before one water exchange occurs. Although the HRE ions are labile structures, Raman spectroscopy probes the actual structure of these octahydrate species. (It is worth mentioning that the intramolecular bond exchange rate is only a few picoseconds, much faster than the water exchange reaction, therefore for such labile structures for instance [Cu(OH_2_)_5_]^2+^ [27], Raman observes an average structure and, so, a single broad mode appears as a result and at higher peak positions than for comparable divalent metal ions.)

### 3.2. YbCl_3_ Solutions 

From thermodynamic measurements, it is known that Lu^3+^ and Yb^3+^ form weak chloro-complexes/ion-pairs [29,30,31,32]. As a model system for the HRE ion hydrates, aqueous YbCl_3_ solutions were investigated. The polarized, depolarized and isotropic Raman scattering profiles of a 3.224 mol·L^−1^ YbCl_3_ solution compared to a solution at 0.802 mol·L^−1^ are presented in Figure 5. Furthermore, the isotropic scattering profiles of three YbCl_3_(aq) solutions at 3.224 mol·L^−1^ (R_w_ = 15.64), 1.600 mol·L^−1^ (R_w_ = 33.06) and 0.802 mol·L^−1^ (R_w_ = 67.55) from 55–700 cm^−1^ are given in Figure 6. Two YbCl_3_ solutions in heavy water were also investigated at 0.422 mol·L^−1^ and at 0.844 mol·L^−1^ and the overview Raman scattering spectra of a 0.422 mol·L^−1^ YbCl_3_ solution in D_2_O is given in Appendix A. In dilute YbCl_3_(aq), the Yb^3+^ ion is fully hydrated indicated by the ν_1_Yb–OH_2_ mode at 394 cm^−1^ for the [Yb(OH_2_)_8_]^3+^ species while in dilute YbCl_3_ solution in heavy water at 0.422 mol·L^−1^ (see Appendix A), the ν_1_Yb–OD_2_ mode of [Yb(OD_2_)_8_]^3+^ appears at 376 cm^−1^ due to the vibrational isotope effect (see discussion further above; H_2_O/D_2_O are considered point masses). 

The ν_1_ Yb–OH_2_ stretching mode in the 3.224 mol·L^−1^ YbCl_3_(aq) solution, with a mole ratio of solute to water at 1 to 15.64 appears at 389 cm^−1^ and shifts with dilution to higher frequencies (see Figure 5). In a 0.400 mol·L^−1^ (R_w_ = 136.98) YbCl_3_(aq) solution the ν_1_ Yb–OH_2_ breathing mode appears at 394 cm^−1^ with a fwhh at 52 cm^−1^ and these band parameters are comparable to the ones in a dilute Yb(ClO_4_)_3_(aq) solution in which the fully hydrated [Yb(OH_2_)_8_]^3+^ exists.

A broad isotropic component at 206 cm^−1^ and a broad feature at 256 cm^−1^ are also observed. The band at 256 cm^−1^ is due to the partially hydrated water molecules of the [Yb(OH_2_)_7_Cl]^2+^ species. A Yb-Cl stretching mode should appear at much higher frequencies, namely at ~500 cm^−1^, but may be very broad and weak and could not be observed (see spectroscopic and DFT results on ZnCl_2_(aq) [33]). The isotropic component at 206 cm^−1^ is assigned to the restricted translation band of water of its O-H∙∙∙O/Cl^−^ units. These findings are evidence that Cl^−^ substitutes water from the first hydration shell of Yb^3+^ and a partially hydrated Yb^3+^- chloro-complex formulated as [Yb(OH_2_)_7_Cl]^2+^ is formed.

The integrated band intensities of ν_1_YbO_8_ band of the fully hydrated species, [Yb(OH_2_)_8_]^3+^, as a function of concentration was determined from quantitative Raman analysis and it turned out that the integrated band intensity, A_394_, does not increase linearly with the total YbCl_3_ concentration (C_T_). However, a linear increase in band intensity would be expected if the Yb^3+^ - octahydrate is the only stable species in YbCl_3_ solution and such a linear relationship was observed in Yb(ClO_4_)_3_(aq) solutions (see Appendix A). The measured integrated band intensity of the ν_1_ YbO_8_ band in YbCl_3_ (aq), A_394_, follows a linear relationship between A_394_ and C_T_ up to ~0.4 mol·L^−1^ but then levels off noticeably at higher YbCl_3_ concentrations (Appendix A). Obviously, above ~0.4 mol·L^−1^ YbCl_3_ fractions of the fully hydrated Yb^3+^ (aq) are converted to a 1:1 Yb^3+^ chloro-complex species. The existence of higher chloro complexes than 1:1 can be convincingly ruled out taking into account the results of earlier anion exchange studies on aqueous rare earth chloride systems [32]. The mole fractions of both species are plotted in Figure 7. The fraction of the chloro-complex at 29%, in the most concentrated solution, is rather small and the fully hydrated species at 71% is still dominant. With dilution, the fraction of the chloro-complex species diminishes quickly and at ~0.4 mol·L^−1^ it is zero.

Shown (Figure 5) is the ν_1_YbO_8_ symmetric stretching mode at 394 cm^−1^ in a 0.802 mol·L^−1^ solution compared to the one at 389 cm^−1^ in a 3.224 mol·L^−1^ solution. An additional isotropic band appears at 256 cm^−1^ in the 3.224 mol·L^−1^ solution which is due to the stretching mode of the chloro-complex species, [Yb(OH_2_)_7_Cl]^2+^ (details in R_iso_ scattering in the terahertz region see Figure 6). The remaining bands in both panels are due to the water being strongly influenced by the solute at the most concentrated solution. First, in the terahertz region (R_VV_ scattering), weak, broad bands appear at 186 cm^−1^ in the 0.802 mol·L^−1^ solution and at 202 cm^−1^ in the 3.224 mol·L^−1^ solution assigned to the restricted translational band of water of the O-H∙∙∙O/Cl^−^ units. Second, very broad bands (R_VV_ scattering) with peak maxima which appear at 712 cm^−1^ (0.802 mol·L^−1^) and 684 cm^−1^ (3.224 mol·L^−1^) are due to the librational bands of water. Third, the band at 1272 (0.802 mol·L^−1^) and 1204 cm^−1^ (3.224 mol·L^−1^) are due to overtones of water librations. Finally, the bands at 1645 and 1647 cm^−1^ respectively are due to the deformation mode of water, ν_2_ H_2_O. 

The formation of a 1:1 complex with Cl^−^ at higher YbCl_3_ concentrations may be written as:(5)$$[\mathrm{Yb}({\mathrm{OH}}_{2}{)}_{8}]{}^{3+}\;+\;{\mathrm{Cl}}^{-}\;⇌\;[\mathrm{Yb}({\mathrm{OH}}_{2}{)}_{7}\mathrm{Cl}]{}^{2+}\;+\;H_{2}O$$

The formation constant for the 1:1 Yb^3+^-chloro-complex, K_1_, may be formulated according to Equation (6):
(6)$$K_{1}=\frac{{[\mathrm{YbCl}}^{2+}]}{{[\mathrm{Yb}}^{3+}{][\mathrm{Cl}}^{-}]}\cdot \frac{f_{{\mathrm{YbCl}}_{2}{}^{2+}}}{f_{{\mathrm{Yb}}^{3+}}\cdot f_{{\mathrm{Cl}}^{-}}}$$
with K_1_′ the “concentration quotient” we get: $K_{1}{=K}_{1}{}^{\prime }\cdot \frac{f_{{\mathrm{YbCl}}_{2}{}^{2+}}}{f_{{\mathrm{Yb}}^{3+}}\cdot f_{{\mathrm{Cl}}^{-}}}$

The concentration quotient can be measured by Raman spectroscopy according to equation (7):(7)$${K_{1}}^{\prime }\;=\frac{{(C}_{T}{-[\mathrm{Yb}}^{3+}])}{{[\mathrm{Yb}}^{3+}]\cdot {[\mathrm{Cl}}^{-}]},$$
where C_T_ is the total YbCl_3_ concentration and the concentrations in brackets denote the equilibrium concentrations of the fully hydrated Yb^3+^ and Cl^−^. The equilibrium concentration of Yb^3+^ determined by Raman spectroscopy allows us to calculate ${K_{1}}^{\prime }$.

The estimated K_1_ value for chloro complex formation in YbCl_3_(aq) from ${K_{1}}^{\prime }$ (see ref [17] for details) equal to ca. 0.06 ± 0.015 and a logK_1_ value at ca. −1.22 follows at 22 °C. (Quantitative Raman spectroscopy applied to these solution spectra with weak and broad low frequency bands is not very precise and therefore a higher uncertainty results.) Data from thermodynamic and spectroscopic studies on YbCl_3_(aq) solutions confirm the weak nature of the complex species [29,30,31,32]. The results on aqueous LuCl_3_ solutions and similar rare earth systems [17,29,30,31,32] confirm our findings on YbCl_3_ solutions. The chloride ion substitutes a water molecule from the flexible first hydration shell of Lu^3+^ and Yb^3+^. With dilution, the weak chloro-complex species dissociates and fully hydrated Yb^3+^(aq) ions detected. This is in contrast to AlCl_3_(aq) solutions, even in concentrated AlCl_3_(aq), Cl^−^ does not substitute water in the first hydration shell of Al^3+^ and it is known that the hydration shell of [Al(OH_2_)_6_]^3+^ is quite inert [22,23]. 

The results of an extensive EXAFS study by Allen and co-workers [34] on 0.1 and 0.01 mol·L^−1^ Lu^3+^-, Yb^3+^ and Tm^3+^- solutions in 0.20 mol·L^−1^ HCl and with 14 mol·L^−1^ LiCl are worthwhile to consider. It could be shown that in solutions with low chloride concentrations, the Ln–O bond distance for Yb^3+^ is consistent with the fully hydrated Yb^3+^. In solutions with an excess of LiCl, it was demonstrated that inner sphere chloro-complexation takes place together with a loss of water [34]. Furthermore, a current study in the terahertz frequency range of YbCl_3_ solutions using FT-IR spectroscopy [35] confirmed weak contact ion pairs as do our recent Raman results on LuCl_3_(aq) [17]. Choppin and Unrein [36] claimed that only outer-sphere ions pairs exist in lanthanide chloride solutions, but such a view has been questioned [17,29,30,31,32]. 

To summarize, the [Yb(OH_2_)_7_Cl]^2+^ modes in chloride solutions could be detected and formation of weak chloro-complexes with Yb^3+^ verified. In dilute solutions (C_T_ < 0.4 mol·L^−1^) the chloro-complex species disappeared upon dilution and [Yb(OH_2_)_8_]^3+^ and Cl^−^ (aq) formed. The chloro-complex formation may be one reason for the data scatter of the recently published Yb–O bond distances and coordination numbers presented for Yb^3+^(aq) and other rare earth chloride systems [35,36]. In recent experimental structural studies, it was observed that inner-sphere chloro-complex species are formed in aqueous LnCl_3_ solution (Ln = Lu and Yb) with high chloride concentrations while in dilute solutions, fully hydrated ions exist [17,34,35].

## 4. Conclusions

Raman measurements on dilute aqueous Lu(ClO_4_)_3_, Yb(ClO_4_)_3_, Tm(ClO_4_)_3_, Er(ClO_4_)_3_ and [Ho(H_2_O)_8_]^3+^ solutions have been carried out. In these solutions, strongly polarized modes at 396 cm^−1^, 394 cm^−1^, 391 cm^−1^, 389 cm^−1^ and 387 cm^−1^ were detected and assigned to the breathing modes, ν_1_ Ln–O of the octahydrates [Lu(H_2_O)_8_]^3+^, [Yb(H_2_O)_8_]^3+^, [Tm(H_2_O)_8_]^3+^, [Er(H_2_O)_8_]^3+^ and [Ho(H_2_O)_8_]^3+^,respectively. The force constants of these Ln–O breathing modes were calculated from these vibrational bands. In dilute perchlorate solutions, these species represent the fully hydrated [Ln(OH_2_)_8_]^3+^ ions. The calculated force constants of the octahydrates of [Lu(H_2_O)_8_]^3+^, [Yb(H_2_O)_8_]^3+^, [Tm(H_2_O)_8_]^3+^, [Er(H_2_O)_8_]^3+^ and, [Ho(H_2_O)_8_]^3+^ strengthen with the corresponding decrease of their Ln–O bond distances. The Ln–O bond of the five octahydrates is not very polarizable and therefore the scattering intensities of the ν_1_ Ln–O bands are small. In Yb(ClO_4_)_3_ in heavy water, the Yb–O breathing mode shifts to 375 cm^−1^ for the deuterated species [Yb(OD_2_)_8_]^3+^ as the result of the vibrational isotope effect changing from H_2_O to D_2_O. 

As a representative example for the lanthanide perchlorate solutions, higher concentrated aqueous Yb(ClO_4_)_3_ solutions were studied and in solutions > 2 mol·L^−1^ a small fraction of contact ion pairs between Yb^3+^ and ClO_4_^−^ were detected. This is supported by the parameters of ClO_4_^−^-bands (peak position, half width, band shapes) as well as changes of the band parameters of the ν_1_ Yb–O breathing mode.

In YbCl_3_ solutions, Cl^−^ penetrates into the first hydration sphere of Yb^3+^(aq) by pushing out a water molecule, and a weak 1:1 chloro-complex species forms. However, the fraction of the chloro-complex diminishes rapidly upon dilution and at a concentration < 0.4 mol·L^−1^, the chloro-complex species vanished. Our Raman spectroscopic findings were substantiated by recently published EXAFS and terahertz FT-IR and results [34,35].

## Figures and Tables

**Figure 1 molecules-24-01953-f001:**
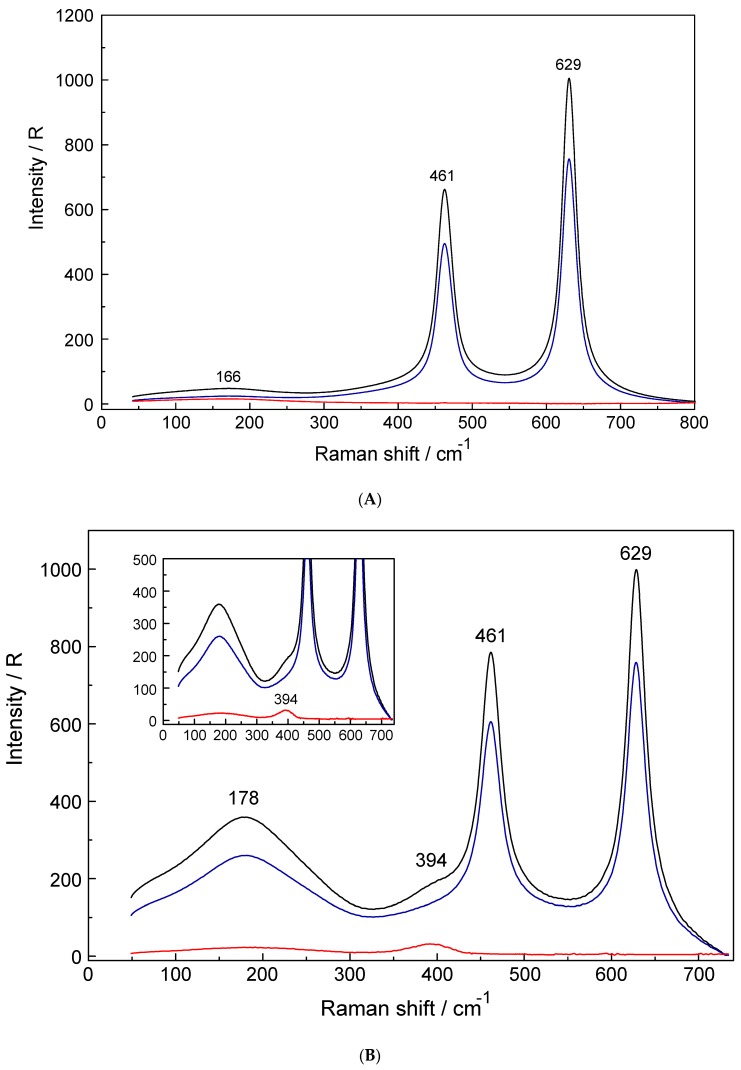
(**A**). Raman scattering profiles in R-format (from top to bottom: R_VV_ (black), R_VH_ (blue) and R_iso_(red)) of a 3.801 mol·L^−1^ NaClO_4_(aq) solution (R_W_ = 12.05). Note the broad weak mode at 166 cm^−1^ is due to the restricted translation of the O-H···H unit of the water molecules. The much larger, depolarized modes at 461 cm^−1^ and 629 cm^−1^ are the deformation modes of perchlorate, ClO_4_^−^(aq). (**B**). A Yb(ClO_4_)_3_ solution spectrum at 0.240 mol·L^−1^ (R_w_ = 226.6) in R-format (spectra from top to bottom: R_VV_ (black), R_VH_ (blue) and R_iso_ (red)). The inset shows the R_iso_ spectrum in greater detail. Note the broad and weak ν_1_YbO_8_ stretching mode at 394 cm^−1^ (fwhh = 52 cm^−1^) of the [Yb(OH_2_)_8_]^3+^ species. The much larger, depolarized bands at 461 cm^−1^ and 629 cm^−1^ are the deformation modes of perchlorate, ClO_4_^−^(aq). (**C**). Raman scattering profiles of a Yb(ClO_4_)_3_ solution at 2.423 mol·L^−1^ in R-format (spectra from top to bottom: R_VV_ (black), R_VH_ (blue) and R_iso_ (red)). The inset shows the R_iso_ spectrum in greater detail. Note the broad and weak ν_1_ YbO_8_ stretching mode at 390 cm^−1^ of the [Yb(OH_2_)_8_]^3+^ species. The much larger, depolarized bands at 461 cm^−1^ and 629 cm^−1^ are the deformation modes of perchlorate, ClO_4_^−^(aq).

**Figure 2 molecules-24-01953-f002:**
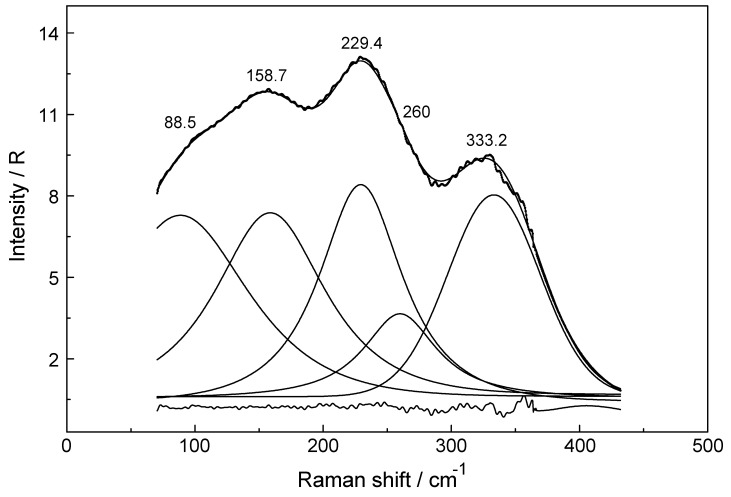
Result of a band fit of an anisotropic Raman scattering profile of a Yb(ClO_4_)_3_ solution at 2.423 mol·L^−1^ in R-format. Shown are the measured spectrum, baseline corrected, the sum curve of the band fit and the band components. Underneath is the residue curve, which is the difference of the measured spectrum and the sum curve.

**Figure 3 molecules-24-01953-f003:**
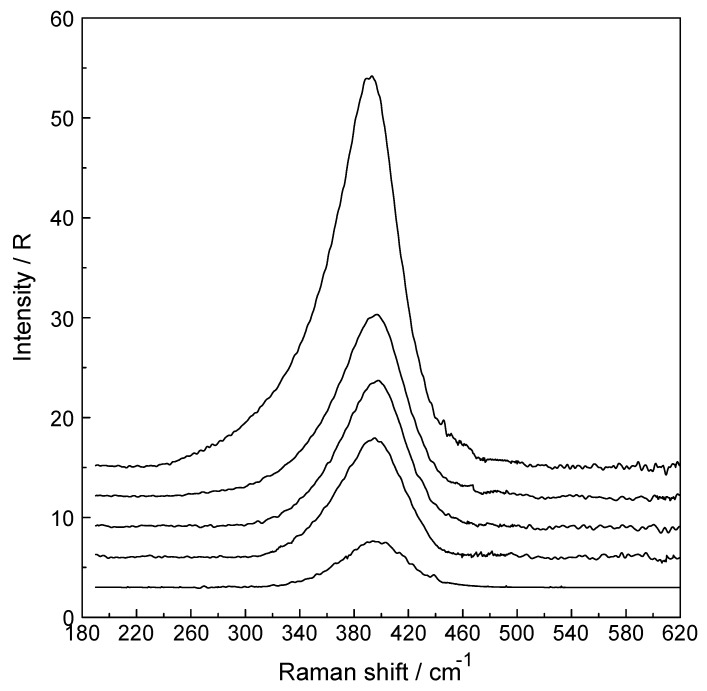
Stack plot of five isotropic Raman profiles in R-format of aqueous Yb(ClO_4_)_3_ solution. From bottom to top: at 0.240 mol·L^−1^, 0.603 mol·L^−1^, 0.808 mol·L^−1^, 1.217 mol·L^−1^ and 2.423 mol·L^−1^.

**Figure 4 molecules-24-01953-f004:**
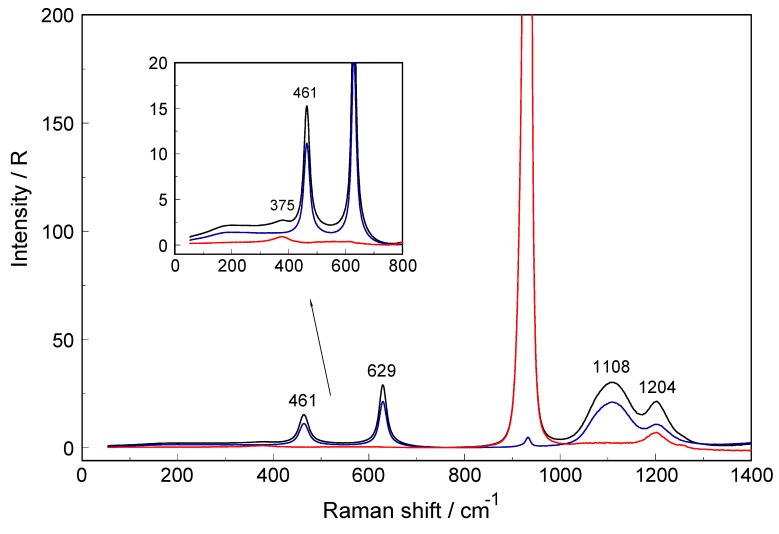
Raman scattering profiles in R-format (spectra from top to bottom: R_VV_, R_VH_ and R_iso_) of a 0.779 mol·L^−1^ Yb(ClO_4_)_3_ solution in D_2_O. The inset shows the low frequency region in larger detail. The weak band at 375 cm^−1^ is assigned to the Yb–OD_2_ mode of the YbO_8_ skeleton which is shifted due to the isotope effect by changing from H_2_O to D_2_O (see also Figure 1B). Note the band at 1204 cm^−1^ which is due to the deformation mode of D_2_O.

**Figure 5 molecules-24-01953-f005:**
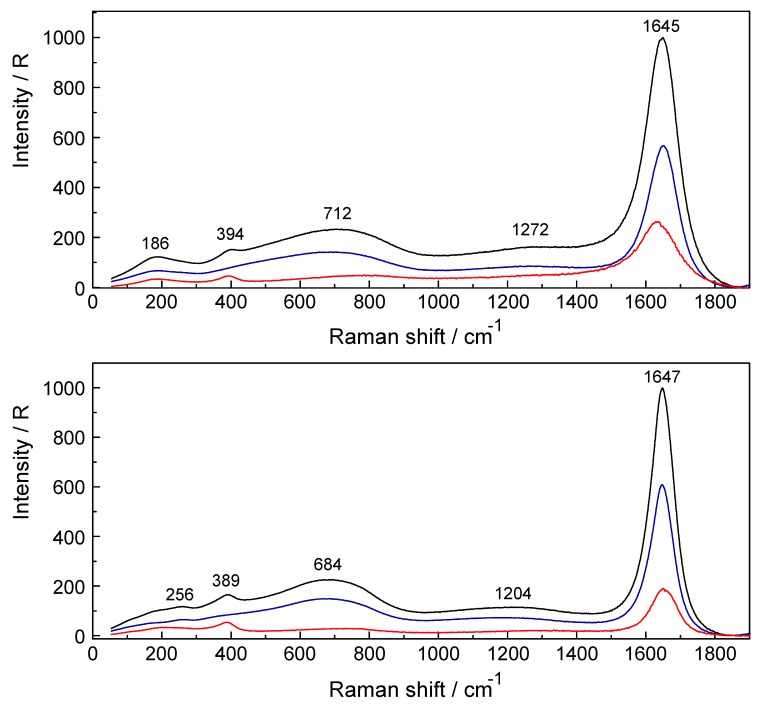
Raman scattering profiles (R_VV_ (black), R_VH_ (blue) and R_iso_ (red)) of two YbCl_3_(aq) solutions from 55–1910 cm^−1^. Top panel: 0.802 mol·L^−1^; Bottom panel: 3.224 mol·L^−1^.

**Figure 6 molecules-24-01953-f006:**
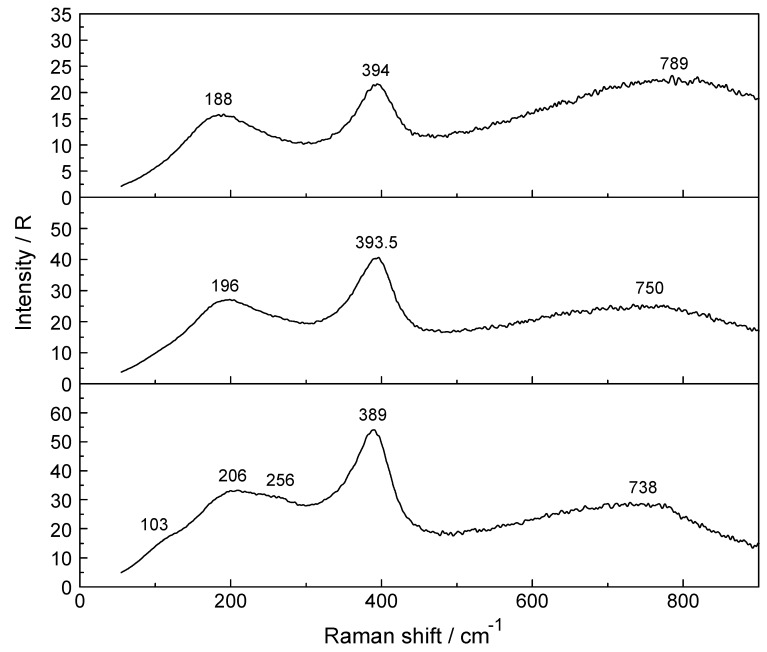
Isotropic Raman scattering profiles of YbCl_3_(aq) solutions from 55–900 cm^−1^. From bottom to top: 3.224 mol·L^−1^, 1.600 mol·L^−1^ and 0.802 mol·L^−1^. The symmetric YbO_8_ stretching mode appears at 389 cm^−1^ in a solution at 3.224 mol·L^−1^ and is shifted to 393.5 cm^−1^ in a 0.802 mol·L^−1^ solution. The band at 256 cm^−1^ is due to the stretching mode of the chloro-complex species, [Yb(OH_2_)_7_Cl]^2+^. The restricted translation band of the O-H···O/Cl^−^ band of water shifts from 206 cm^−1^ in the 3.224 mol·L^−1^ solution and appears at 188 cm^−1^ in the 0.802 mol·L^−1^ solution. The broad librational band of water shifts from 738 cm^−1^ to 789 cm^−1^ in going from a 3.224 mol·L^−1^ solution to the one at 0.802 mol·L^−1^.

**Figure 7 molecules-24-01953-f007:**
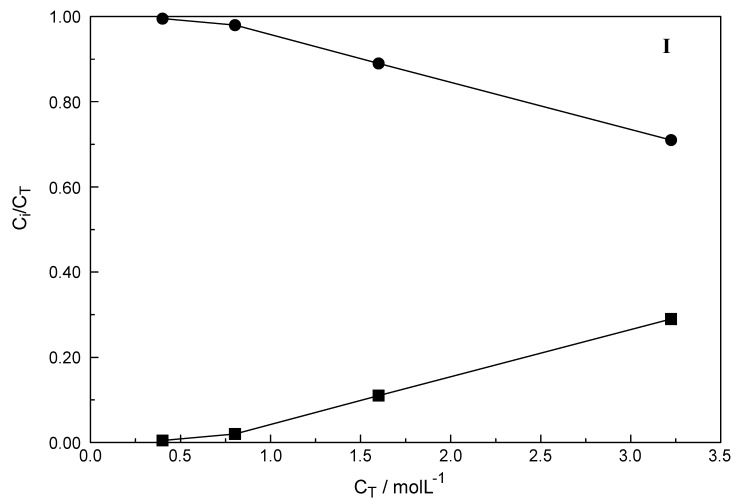
Fraction of species detected by quantitative Raman spectroscopy. The filled circles denote the [Yb(OH_2_)_8_]^3+^, the fully hydrated Yb^3+^ and the filled squares the mono chloro-complex species, [Yb(OH_2_)_7_Cl]^2+^. Note the error bar in the right corner of the graph.

**Table 1 molecules-24-01953-t001:** Band parameters such as peak positions, ν_1_, force constants*, k_Ln–O_,* the full width at half heights, and the scattering intensities for the Ln–O breathing mode of Ho^3+^, Er^3+^, Tm^3+^, Yb^3+^ and Lu^3+^ for the [Ln(OH_2_)_8_]^3+^ species at 22 °C. Furthermore, the literature data for the water exchange [24] and the Ln–O bond distances [7] for these [Ln(OH_2_)_8_]^3+^ species are given.

[Ln(OH_2_)_8_]^3+^	ν_1_ Ln–O /cm^−1^	*k_Ln–O_*/Nm^−1^	Fwhh/cm^−1^	S_h_	$k_{ex}^{298}/10^{8}\;s{-}^{1}\;a)$	τ_w_/ns	Ln–O/Å b)
Ho^3+^	387	158.97	55	0.0170	1.91	5.24	2.379
Er^3+^	389	160.62	54	0.0168	1.18	8.48	2.364
Tm^3+^	391	162.27	53	0.0165	0.81	12.34	2.350
Yb^3+^	394	164.77	52	0.0160	0.41	24.39	2.324
Lu^3+^	396	166.45	52	0.0156	-	-	2.316

a) Ref. [24]; ^−^ data at 25 °C. b) Ref. [7]; EXAFS data from L_3_ edge (one shell fit) on aqueous trifluoromethansulfonate solutions.

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
