# Peer review of "On the Hydration of Heavy Rare Earth Ions: Ho3+, Er3+, Tm3+, Yb3+ and Lu3+—A Raman Study"

_molecules, 2019, doi:10.3390/molecules24101953_

Round 1

Author Response

Reviewers comments:

In the manuscript, the authors study the Raman spectra of aqueous heavy rare earth (HRE) ions to understand their hydration process. In their analysis, the regular Raman spectra are reduced to obtain R-format spectra, which have improved data quality in the low wavenumber region. The formation of the HRE ion octahydrates can be confirmed from the observation of a ~390 cm-1 Raman peak corresponding to its breathing mode. Furthermore, as the HRE ion changed between Lu3+, Yb3+, Tm3+, Er3+, and Ho3+, the peak also red shifts reflecting the increased bond length. The position of this breathing mode also shows concentration dependence, which is attributed to ion pair formation. In concentrated YbCl3 aqueous solution, an extra mode at 256 cm-1  is detected, which comes from the chloro- complex species.

The major findings in the draft are well supported by the experimental evidences. The authors also show adequate analysis of their data. Hence, the conclusions drawn here seem convincing. Given that I see no problem publishing this manuscript, some issues must be addressed, and changes need to be made:

1.  Even though the overall logic in this paper is clear, some of the language seems redundant. This paper would be more reader friendly if authors could rewrite some paragraphs in a more concise way. For example, on page 16, the “weak complexes” between Yb and Cl- have been “shown” multiple times. Also, on the same page, a sentence like “in dilute solutions the chloro-complex species disappeared upon dilution” makes the draft unnecessarily wordy.

2.  Some key quotes in the draft are wrong or mislabeled. On page 15, the last line, the “see Figure S9” should actually be “Figure S11”. On page 16, “according to equation (7)” is confusing, because there is no equation (7) in the paper.

3.  Quite some typos can be found in the draft as well. Please double check before final publishing.

4.  In the section about YbCl3 concentration, figure 5 shows 3.2, 1.6 and 0.8 mol/L while figure 4 only shows 3.2 and 0.8 mol/L.

5.  Figure S3 gives the fitting of a spectrum. However, in the main text, there is no correspondent spectrum that resembles this piece of spectrum.

Reply to Reviewers comments:

Reviewer 1:

1.      We agree with the reviewer´s statement and we improved the section on YCl3(aq). First, we incorporated two new Figures; Figure 7 and Figure S11. In Figure 7 we show: Fraction of species detected by quantitative Raman spectroscopy. The filled circles denote the [Yb(OH2)8]3+, the fully hydrated Yb3+ and the filled squares the mono chloro-complex species, [Yb(OH2)7Cl]2+. Note the error bar in the right corner of the graph. Such a graph should clarify the quantitative results measured by Raman in YbCl3 solutions better. Second, we introduced Figure S11 to give the equilibrium concentrations of [(Yb(OH2)8)3+] (brackets denote equilibrium concentrations). The linear dependence of the integrated band intensity, A394 as a function of the Yb(ClO4)3 concentration with A394=1303.7*CT (see Figure S3). The lower curve shows the integrated band intensity of the band at 394 cm-1 in YbCl3 solutions (black squares). The solute concentration is denoted as CT. From this graph the [(Yb(OH2)8)3+] follows directly.

2.      We have fixed the quotes and labels. We have introduced two new equations (6 and 7) to define the K1 value better. See text in manuscript 421 pp.

3.      We have double checked for typos. 

4.      We have fixed the problem with Figures 4 and 5 which were inadvertently reversed. These are now Figures 5 and 6..

5.      We agree that the results of the band fit of the anisotropic spectrum were only mentioned briefly. So, we expanded the discussion and we presented a Table S1 to summarize the    band fit results and we moved the former Fig. S3 (band fit of anisotropic scattering) to the main text which is now Figure 2 at line 234 pp. Text follow at lines 209 pp.

Reviewer 2 Report

See attachment

Author Response

Review 2:

Raman spectra of aqueous Ho3+ , Er3+ , Tm3+ , Yb3+  and Lu3+  perchlorate solutions were measured, and in the terahertz region, strongly polarized Raman bands were detected at 387 cm-1, 389 cm-1, 391 cm-1, 394 cm-1, and 396 cm-1, respectivelywhich were assigned to the symmetric Ln-O stretching modes of

the octahydrates [Ho(H2O)8]3+, [Er(H2O)8]3+,    [Tm(H2O)8]3+, [Yb(H2O)8]3+,    and [Lu(H2O)8]3+.   This

paper is one of a series studies on Raman spectra of aqueous rare earth ions Ln3+ by this group. This paper is publishable; however, some mistakes and descriptions should be further revised.

1, It is better to illustrate Raman spectra of Ln3+ in one picture to demonstrate the band shift trend since it is extremely tricking to identify  2 cm-1 difference with  52 cm-1  width at half height.

2, What is DFT calculated symmetric Ln-O stretching modes of [Ln(H2O)8]3+? The comparison of measured and calculated frequencies is very helpful for the band assignment.

3, A broad isotropic component at 206 cm-1 and an unresolved broad feature at 256 cm-1 are also observed. These findings are clear evidence that Cl- has penetrated into the first hydration shell of Yb3+ and the partially hydrated Yb3+ chloro-complex formulated as [Yb(H2O)7Cl]2+. How to know how many Cl irons to penetrate into shell?

Reply to reviewer 2:

1.)    We do not think that a special graph is necessary to show the breathing modes of the heavy rare earth ions in water because the perchlorate bands of the deformation modes serve as a wavenumber internal standard and allow determination of the peak positions of these broad bands with an accuracy at ± 1 cm-1. We included a short statement in the Experimental Section. See lines 136 pp.

2.)    In our recent paper on Lu3+(aq) solutions, we carried out DFT calculations of the [Lu(OH2)8]3+. As it turns out, these calculations are very much basis set dependent and may not serve as reliable values. See also our discussion at line 321 pp. We would like to refer for instance to results on the basis set dependence of the frequencies on Ce3+ nona-hydrate clusters by Clark (Clark, A.E., Density Functional and Basis Set Dependence of Hydrated Ln(III) Properties. J. Chem. Theory Comput. 2008, 4, 708–718, doi:10.1021/ct700317p.). We have discussed this problem in our recent paper (Rudolph and Irmer; Molecules 2018, 23, 3237; doi:10.3390/molecules23123237). Furthermore, theoretical data on the breathing modes of these heavy rare earth ions were already published extensively in ref. 13 of our manuscript in order to demonstrate the problems connected with theoretical frequencies on rare earth ion hydrates.

3.)    The chloro complex species exist only in concentrated solutions and a mono chloro complex is the most likely species. Such an assumption is also supported by data from other studies (see for instance ref. 33 and 17 in our manuscript). The band assignments, however, follow similar studies from our recent work. The integrated band areas of the fully hydrated Yb3+(aq)  in the YbCl3(aq) solutions do not grow linearly but curve towards smaller values. Therefore, we have given a graph in Figure S11 to show this. In Figure 7, we have inserted an additional graph (Figure 7) giving the fractions of the species, fully hydrated Yb3+ and Yb-Cl complex. See also text from line 375. This is a sign that a chloro complex species (as opposed to an outer-sphere ion pair) must have formed. We have explained our quantitative Raman procedure in our recent paper on LuCl3(aq) solutions (Rudolph and Irmer; Molecules 2018, 23, 3237; doi:10.3390/molecules23123237).

This manuscript is a resubmission of an earlier submission. The following is a list of the peer review reports and author responses from that submission.